# Child support grant expansion and cognitive function among women in rural South Africa: Findings from a natural experiment in the HAALSI cohort

Rishika Chakraborty[1]*, Lindsay C. Kobayashi[2], Janet Jock[3], Coady Wing[3], Xiwei Chen[4], Meredith Phillips[4], Lisa Berkman[5], Kathleen Kahn[6,7], Chodziwadziwa Whiteson Kabudula[6], Molly Rosenberg[4,6]

1 Department of Environmental and Occupational Health, Indiana University School of Public Health-Bloomington, Bloomington, Indiana, United States of America, 2 Department of Epidemiology, University of Michigan School of Public Health, Ann Arbor, Michigan, United States of America, 3 O'Neill School of Public and Environmental Affairs, Indiana University-Bloomington, Bloomington, Indiana, United States of America, 4 Department of Epidemiology and Biostatistics, Indiana University School of Public Health-Bloomington, Bloomington, Indiana, United States of America, 5 Harvard Center for Population and Development Studies, Cambridge, Massachusetts, United States of America, 6 MRC/Wits Rural Public Health and Health Transitions Research Unit (Agincourt), School of Public Health, Faculty of Health Sciences, University of the Witwatersrand, Johannesburg, South Africa, 7 Department of Epidemiology and Global Health, Umeå University, Umeå, Sweden

* rchakra@iu.edu

**Data Availability Statement:** The HAALSI data used in this article are publicly available in the Harvard Dataverse. Data from the "HAALSI

## Abstract

### Background

Cash transfers are a promising but understudied intervention that may protect cognitive function in adults. Although South Africa has a rapidly ageing population, little is known about the nature of association between cash transfers and cognitive function in this setting.

### Objectives

We leveraged age-eligibility expansions to South Africa's Child Support Grant (CSG) to investigate the association between duration of CSG eligibility and cognitive function of biological mothers of child beneficiaries in South Africa.

### Methods

We analysed 2014/2015 baseline data from 944 women, aged 40–59 years with at least one CSG-eligible child, enrolled in the population-representative HAALSI cohort in Agincourt, South Africa. Duration of CSG eligibility for each mother was calculated based on the birth dates of all their children and the CSG age-eligibility expansion years (2003–2012). Cognitive function was measured using a cognitive battery administered at the HAALSI baseline interview. Linear regression was used to estimate the association between duration of CSG eligibility, dichotomized as low ($\leq$10 years) and high (>10 years) eligibility, and cognitive function z-scores of the mothers.

Baseline Survey" and the "HAALSI Dementia Wave 1" can be found here https://dataverse.harvard.edu/dataverse/harvard?q=HAALSI.

**Funding:** This current work was supported by U.S. National Institute on Aging (grant number 1R01AG069128) to MR and LCK; the HAALSI study was supported by the U.S. National Institute on Aging (grant number P01 AG041710) to LB. HAALSI is nested within the Agincourt Health and Demographic Surveillance System site, which is supported by the University of the Witwatersrand and Medical Research Council, South Africa, and the Wellcome Trust, UK (grant numbers 058893/Z/99/A; 069683/Z/02/Z; 085477/Z/08/Z; 085477/B/08/Z) to KK. The funders had no role in study design, data collection and analysis, decision to publish, or preparation of the manuscript.

**Competing interests:** The authors have declared that no competing interests exist.

## Results

High vs. low duration of CSG eligibility, was associated with higher cognitive function z-scores in the full sample [β: 0.15 SD units; 95% CI: 0.04, 0.26; p-value = 0.01]. In mothers with one to four lifetime children, but not five or more, high vs. low duration of CSG eligibility, was associated with higher cognitive function z-scores [β: 0.19 SD units; 95% CI: 0.05, 0.34, p-value = 0.02].

## Conclusion

Government cash transfers given to support raising children may confer substantial protective effects on the subsequent cognitive function of mothers. Further studies are needed to understand how parity may influence this relationship. Our findings bring evidence to policymakers for designing income supplementation programmes to promote healthy cognitive ageing in low-income settings.

## Introduction

In 2019, about 57 million people worldwide were living with dementia and this number is projected to increase to 83 million in 2030, 116 million in 2040 and 153 million by 2050 [1, 2]. Alzheimer's disease and related dementias (ADRD) is one of the leading causes of disabilities, poor health, and decreased quality of life [3]. The burden of ADRD falls disproportionately on people living in low- and middle-income countries, where two-thirds of cases currently occur [4]. Largely due to changes in demographic patterns and population ageing, it is estimated that by 2050 there will be about 7.62 million people living with ADRD in sub-Saharan Africa [5]. Despite this growing public health concern, limited research exists on ADRD risk and prevention in sub-Saharan African countries like South Africa [6, 7].

Prior evidence suggests that socioeconomic conditions throughout the life course shape ADRD risk [8–12]. A diagnosis of ADRD is typically preceded by a prodromal period of at least five years of accelerated decline in cognitive function [13]. Socioeconomic status indicators across the life course such as education, occupation and income have been linked to cognitive outcomes in older adults [14]. These socioeconomic status indicators may be associated with later life cognitive function by providing individuals with assets or tools to attain resources that can promote healthy cognitive ageing [9]. For instance, higher income allows adults to access nutritious food and healthy housing, which can positively influence their later life cognitive outcomes. In contrast, low income or lack of wealth may lead to chronic stress, which in turn can negatively affect their cognitive health [9].

The cumulative presence of socioeconomic advantages and thus cognitively stimulating exposures across the life course may have a protective effect on cognitive function and ADRD risk by promoting cognitive reserve [15, 16]. Cognitive reserve is a theoretical construct that is thought to improve the brain's ability to cope with pathological damage and can be accrued across one's life course [15]. Individuals with higher cognitive reserve may use their pre-existing neural networks more efficiently or may recruit compensatory mechanisms to perform a cognitive task, consequently coping better with brain pathology [17]. Socioeconomic status indicators, such as income, are considered proxy measures for cognitive reserve [18]. However, little is known about the nature of association between income and cognitive function in low-income settings, such as in rural South Africa [6, 19].

South Africa has a rapidly ageing population of people whose lives have been affected by structural racism during Apartheid from 1948 to 1994 [20]. Since the end of Apartheid, South Africa has implemented several social protection programmes, the largest being the Child Support Grant (CSG) [21, 22]. Implemented in 1998, the CSG is an unconditional cash transfer program designed to alleviate child poverty, offset the costs of raising children and improve their health and nutrition by providing financial assistance that supplements household income [23]. While the CSG is intended for the primary caregivers of the child beneficiaries, nearly 90 per cent of recipients are their biological mothers [24, 25]. When the CSG was first rolled out, only children under seven years were eligible. However, between 2003 and 2012, CSG eligibility was progressively expanded over time and, since 2012, children under the age of 18 have been eligible [22]. Prior research has shown that cash transfers like the CSG can raise living standards, improve health outcomes, and empower women within their households and communities [26]. CSG has been associated with several positive outcomes in child beneficiaries including alleviating food insecurity, improving education, school attendance, and anthropometric outcomes [22]. However, few studies have investigated the impacts of CSG transfers on the caregivers of child beneficiaries. One study found that CSG recipient mothers had a higher probability of being in the labor market and being employed [27], while another reported adults in CSG recipient households had improved mental health [28]. Mothers of CSG recipients have also reported having higher financial independence [29], decision making powers [29, 30], lower worry and stress [30], and improved social relationships within households and communities [30].

What remains unclear is whether cash transfers like the CSG can impact cognitive function and ADRD risk among the caregivers of child beneficiaries. Since cash transfers supplement income, they may help accrue cognitive reserve, however, few studies have investigated the causal effects of cash transfers on ADRD risk. Only one study in Mexico [31], one in United States [32], and another in South Africa [33] used natural experiments to demonstrate the causal effects of cash transfers in improving cognitive function in older adults. These studies found that higher social protection income led to substantial improvements in memory and other domains of cognitive function in older adults [31, 32]. Thus, the causal relationship between cash transfers and mid-to-later life cognitive function in adults from low-resource settings such as rural South Africa remain understudied.

We thus aimed to exploit the exogenous variation in CSG eligibility, due to iterative age eligibility expansion policies, as a natural experiment to estimate the causal effect of cash transfers on cognitive function among biological mothers in South Africa. Based on the cognitive reserve theory, we hypothesized that a greater duration of CSG eligibility would improve midlife cognitive function among mothers, since such exposure increases income, thereby promoting cognitive reserve. This study will contribute to the literature linking social protection policies and cognitive function in an underrepresented and rapidly ageing population in a middle-income country.

## Materials and methods

### Study setting and design

This study is set in the rural Agincourt area of Bushbuckridge sub-district in Mpumalanga province, South Africa. Since 1992, the study area, currently consisting of 31 villages and approximately 116,000 inhabitants, has been under continuous surveillance through the Agincourt Health and Socio-Demographic Surveillance System (HDSS) run by the Medical Research Council/Wits University Rural Public Health and Health Transitions Research Unit [20, 34]. While the socioeconomic situation of the Agincourt population has

improved post-Apartheid, the region still faces difficulties in accessing basic services, such as electricity and piped water [20]. This population is Human Immunodeficiency Virus hyperendemic (~20 per cent prevalence in 2010) [35], and has high unemployment rates [20].

Cash transfers from social protection programmes, such as the CSG, are an important source of household income in the study region [20]. Primary caregivers of age-eligible children whose income falls below a means-tested level (about 10 times the grant amount for single caregivers) are eligible to apply for the CSG [22, 25]. When the CSG was first introduced, uptake rates were low, with only 28 per cent of CSG-eligible households in Agincourt submitting a CSG application in 2001 [25]. However, uptake rates and eligibility have increased over time and the CSG currently reaches over 1.1 million children in Mpumalanga province every month [36]. Since its implementation in 1998, the age eligibility was expanded from children under seven to under age nine in 2003, under age 11 in 2004, under age 14 in 2005, under age 15 in 2009, and through the age of 16 in 2010. Since 2012, children under the age of 18 have been eligible [21, 22]. The grant payment has also increased over time to match inflation, from 100 Rand (~$7 USD) in 1998 to 510 Rand (~$27 USD) in October 2023 per month for each child [37].

Using these CSG age eligibility expansions as natural experiments, we analysed baseline data collected in the 2014/15 'Health and Ageing in Africa: A Longitudinal Study of an INDEPTH Community in South Africa (HAALSI)' [34]. HAALSI is a longitudinal cohort study of 5059 adults (2345 men and 2714 women) aged ≥40 years, representative of the Agincourt HDSS as its sampling frame (~86 per cent response rate). Trained, local fieldworkers collected household and individual data using computer assisted personal interviews [34]. All interviews were conducted in the local Xitsonga language with interview materials translated from English and back-translated for reliability.

Ethical approval for HAALSI was obtained from the Wits University Human Ethics Committee (#M141159), the Harvard T. H. Chan School of Public Health Office of Human Research Administration (#13–1608), and the Mpumalanga Provincial Research and Ethics Committee. Each HAALSI participant provided written informed consent prior to being interviewed. Participants unable to read had a witness and used inked fingerprint to sign consent form. This current study was deemed to be a 'Not Human Subjects Research' by the Institutional Review Board of Indiana University Bloomington (protocol #2002584956) since the data was de-identified prior to analysis.

## Study sample

Study participants were CSG-eligible women, defined as HAALSI women between the ages of 40 to 59 years who had at least one CSG-eligible biological child. We restricted our sample to only biological mothers (excluding other female relatives/fathers/male relatives) since nearly 90 per cent of primary caregivers receiving the CSG are biological mothers of eligible children [25]. After removing women who did not have any children (n = 224), who did not have CSG-eligible children (n = 1363), and with implausible childbirth ages (≤10 or ≥60 years old) (n = 2), there were 1125 CSG eligible women in the HAALSI cohort. We then excluded women who were ≥ 60 years of age (n = 181) since this group may not be representative of other women in their later-life in this rural South African setting and due to their small sample size. The older women would have been at least 38 years old at the time of childbirth in order to have CSG-eligible children, making their reproductive history fall outside the typical childbearing age which was between 20–34 years in South Africa in 1998 [38]. Therefore, the final analytic sample included 944 CSG-eligible mothers (Fig 1).

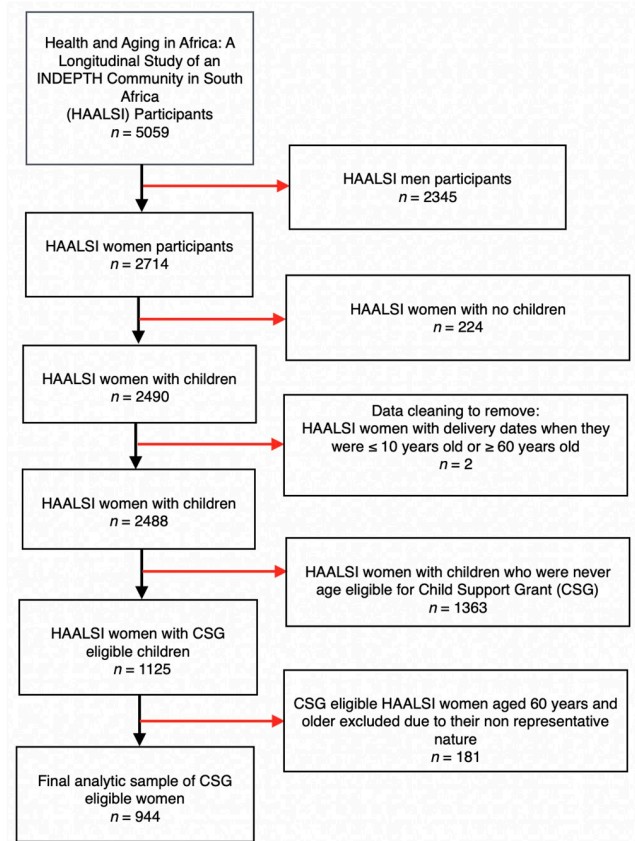

**Fig 1. Flowchart describing the cohort construction of 944 child support grant eligible mothers enrolled in health and ageing in Africa: A longitudinal study of an INDEPTH community in South Africa (HAALSI), 2014/15.**

## Study measures

**Exposure variable: Child Support Grant (CSG) eligibility.** The cumulative duration of CSG eligibility at HAALSI baseline, in years, for each CSG-eligible mother was the exposure of interest. This was calculated based on the birth dates (month and year) of all their children and the CSG age-eligibility expansion years. We aggregated the potential years of CSG eligibility for each child beneficiary to calculate the total duration of CSG eligibility for the mothers. Take, for instance, a woman with two children, one born in January 1994 and the other born in January 1995. Based on their birth dates and the expansions implemented between 2003 and 2012, the child born in 1994 will contribute 7.5 years of CSG eligibility, while the child born in 1995 will contribute 12.5 years of CSG eligibility by January 2016. Thus, for the mother, we would calculate a total CSG eligibility of 20 years at HAALSI baseline. An excel workbook showing our CSG eligibility calculation is included in our Supplementary Materials. Among women of the same age and with similar number of children, the variation in CSG eligibility duration plausibly leads to exogenous differences in CSG exposure over time. In contrast, the actual uptake of CSG benefits may be associated with individual risk factors or early outcomes such as education or high early life cognitive functioning that could confound the relationship between CSG and later outcomes [39]. Thus, in this paper, we focus on the "intent to treat" relationship between exogenous variation in years of CSG eligibility (adjusted for age, education, and number of children) and cognitive functioning.

For current analysis, we categorized the duration of CSG eligibility into a binary variable: ≤10 years ("low") versus >10 years ("high"). This choice of categorisation was informed by a data driven decision making, where we observed a threshold of higher cognitive function scores for mothers who had greater than 10 years of CSG eligibility compared to those who had less. We first mapped the relationship between all CSG eligibility years (as indicator variables) and cognitive function z-scores. To better understand the trend, we then grouped the CSG eligibility years into five-year intervals to visualize the same relationship. We observed that mothers with 1 to 5 years and 5.5 to 10 years of CSG eligibility duration had similar cognitive function scores while those with greater than 10 years had relatively higher cognitive function scores (S3 Fig in S1 File). Given this observation, we decided to use 10 years of CSG eligibility as our threshold for analysis (the CSG variable construction process has been described in further details under Child Support Grant Variable Construction in our Supplementary Materials). We also conducted several sensitivity analyses using CSG as a continuous variable and with different cut-offs for duration of CSG eligibility to assess the robustness of this threshold (Please see Sensitivity Analysis, S1 to S3 Tables in S1 File).

**Outcome variable: Cognitive function.**    A cognitive battery was administered in the HAALSI study interview, which included measures of orientation (ability to correctly state the date, month, year, and South African President; 4 points total) and episodic memory (immediate and delayed recall of ten words that were read out loud by the interviewer; 20 points total) [40]. These measures were selected for use as the inability to orient oneself in time and place and loss of episodic memory are hallmark symptoms of ADRD, and are highly sensitive to ageing-related change [41, 42]. This cognitive battery was adapted from the United States Health and Retirement Study (US HRS) and has been used in other HRS International Partner Studies around the world [43, 44] including South Africa [33, 40, 45]. The cognitive battery also had 2 numeracy measures (forward count and number skip pattern), which we did not include in the present analysis. This is because numeracy is a skill learned through education and given the limited educational opportunities that our study sample faced during Apartheid, those who are not numerate may be particularly disadvantaged for the numeracy tests [40, 41]. For present analysis, we used the continuous z-standardized score of the 24-point composite score (with mean 0 and standard deviation 1) [40].

**Sociodemographic variables.**    The sample of CSG-eligible mothers was described according to their age at the time of the HAALSI baseline interview and CSG implementation (years), number of CSG-eligible children, lifetime number of children (one to four, and five and more), education (none, some primary, and some secondary or more), marital status (never married/ separated/divorced/widowed, and currently married), religion (none, Christianity, and African traditional), country of origin (South Africa, and Mozambique/other), and wealth index quintiles (based on household characteristics and ownership of vehicles, livestock and household items) [34].

## Statistical analysis

CSG-eligible mothers were stratified into two groups according to their number of lifetime children, one to four and five and more children, since the number of lifetime children is structurally linked to higher duration of CSG eligibility. This categorization was chosen to distinguish those with grand multiparity of five and more births [46]. For descriptive analysis, we used Chi-square tests and Mann Whitney U tests for categorical and continuous variables, respectively. Ordinary Least Square regression models were used to estimate the association between duration of CSG eligibility and cognitive function z-score, first in the full sample, and

then stratified by lifetime number of children subgroups. All models were adjusted for age of mother (as a linear and a quadratic term), education (as none, some primary, some secondary or more) and lifetime number of children indicator variables. Statistical analyses were conducted using R statistical software (version 3.6.0, Vienna, Austria). Threshold for significance was defined as p value < 0.05.

## Results

Characteristics of the sample have been described in Table 1. In the full sample, the median duration of CSG eligibility for the mothers was 26.25 years (IQR: 16 to 41 years, Range: 1 to 98.5 years). These years of eligibility could be simultaneously accrued over multiple children, with the median number of CSG eligible children in the full sample being 2 (IQR: 1 to 3 children, Range: 1 to 9 children). The median age of the mothers at the time of the study interview was 50 years (IQR: 44 to 54 years, Range: 40 to 59 years) while at the time of CSG

**Table 1. Sociodemographic characteristics of child support grant eligible mothers according to low (≤10 years) and high (>10 years) duration of child support grant eligibility, health and ageing in Africa: A longitudinal study of an INDEPTH community in South Africa, 2014/15.**

| Characteristics | Overall Sample N = 944 | Low CSG*duration N = 136 | High CSG duration N = 808 | P value |
|---|---|---|---|---|
| Continuous variables [Median (IQR**)] | | | | |
| Number of CSG eligible children | 2 (1, 3) | 1 (1,1) | 3 (2,4) | <0.001 |
| Mother's age in 2014/15 | 50 (44, 54) | 54 (51, 57) | 49 (43, 53) | <0.001 |
| Mother's age in 1998 (CSG implementation) | 34 (28, 38) | 38 (35, 41) | 33 (27, 37) | <0.001 |
| Categorical variables [N (%)] | | | | |
| **Lifetime Number of Children** | | | | |
| 1 to 4 children | 390 (41.3) | 62 (45.6) | 328 (40.6) | 0.27 |
| 5 and more children | 554 (58.7) | 74 (54.4) | 480 (59.4) | |
| **Education** | | | | |
| None | 275 (29.1) | 47 (34.6) | 228 (28.2) | 0.05 |
| Some Primary schooling | 340 (36.0) | 54 (39.7) | 286 (35.4) | |
| Secondary or more schooling | 329 (34.9) | 35 (25.7) | 294 (36.4) | |
| **Marital status** | | | | |
| Currently married | 515 (54.6) | 65 (47.8) | 450 (55.7) | 0.09 |
| Other^ | 429 (45.4) | 71 (52.2) | 358 (44.3) | |
| **Country of origin** | | | | |
| South Africa | 651 (69.0) | 94 (69.1) | 557 (68.9) | 0.98 |
| Mozambique or other | 292 (31.0) | 42 (30.9) | 250 (30.9) | |
| **Religion** | | | | |
| None | 28 (3.0) | 2 (1.5) | 26 (3.2) | 0.17 |
| Christianity | 901 (95.4) | 134 (98.5) | 767 (94.9) | |
| African Traditional | 15 (1.6) | 0 | 15 (1.9) | |
| **Wealth Index** | | | | |
| Quintile 1 (lowest) | 180 (19.1) | 19 (14.0) | 161 (20.0) | 0.36 |
| Quintile 2 | 179 (19.0) | 26 (19.1) | 153 (18.9) | |
| Quintile 3 | 172 (18.2) | 22 (16.1) | 150 (18.6) | |
| Quintile 4 | 192 (20.3) | 33 (24.3) | 159 (19.7) | |
| Quintile 5 (highest) | 221 (23.4) | 36 (26.5) | 185 (22.9) | |

*CSG: Child Support Grant;

**IQR: Inter Quartile Range;

^ never married/separated or divorced/widowed

implementation in 1998, the median age was 34 years (IQR: 28 to 38 years, Range: 24 years to 43 years). Over four-fifths (86 per cent, 808/944) of the full sample had high duration of CSG eligibility (>10 years). About four-fifths (84 per cent, 328/390) of the mothers with one to four lifetime children and 87 per cent (480/554) of the mothers with five and more lifetime children had high duration of CSG eligibility. Approximately one-third of the mothers (~59 per cent, 554/944) had five or more lifetime number of children. Around one-third (~35 per cent, 329/944) of the mothers had received some secondary or higher education and over half (~55 per cent, 515/944) were currently married. About two-thirds of the mothers (69 per cent, 651/944) were originally from South Africa, and majority (95 per cent, 901/944) were Christian (Table 1). Some sociodemographic characteristics significantly differed across the mothers according to their CSG eligibility status (Table 1). Mothers with high duration of CSG eligibility were younger and had more CSG eligible children than mothers with low duration of CSG eligibility. Mothers with high duration of CSG eligibility were marginally more likely to have achieved some secondary or higher education compared to their counterparts with low duration of CSG eligibility. No meaningful differences by duration of CSG eligibility were observed for lifetime number of children, marital status, country of origin, religion, and wealth index.

In the full sample, in unadjusted analysis, high duration of CSG eligibility was associated with higher cognitive function z-scores in the mothers (β = 0.27 SD units, 95 per cent CI: 0.11, 0.43). After adjustment for mother's age, education and lifetime number of children, the magnitude of the observed association was attenuated (adjusted β = 0.15 SD units, 95 per cent CI: 0.04, 0.26; Table 2, Fig 2). In subgroup analyses, high duration of CSG eligibility was associated with higher cognitive function z-scores only among mothers with one to four children (β = 0.37 SD units, 95 per cent CI: 0.12, 0.62). The magnitude of this association was also attenuated after covariate adjustment (adjusted β = 0.19 SD units, 95 per cent CI: 0.05, 0.34; Table 2, Fig 2). In mothers with five or more children, unadjusted analysis indicated that high duration of CSG eligibility was marginally associated with higher cognitive function (β = 0.21 SD units, 95 per cent CI: 0.003, 0.41), but no meaningful associations between duration of CSG eligibility and cognitive function z-scores were observed in adjusted analyses (Table 2).

In our sensitivity analyses, we analysed CSG in three ways: a) as a continuous variable and dichotomized the years of CSG eligibility using two different thresholds near the 10-year cut-off, b) one at 8 years and c) another at 12 years (Supplementary Materials: Sensitivity Analysis).

**Table 2. Estimated effects of high child support grant eligibility (>10 years) compared to low child support grant eligibility (≤10 years) on cognitive function in child support grant-eligible mothers, health and ageing in africa: A longitudinal study of an INDEPTH community in South Africa, 2014/15.**

| Analytic Sample | Beta Coefficient (β) | 95% Confidence Intervals | P value |
|---|---|---|---|
| *Full Sample (n = 944)* | | | |
| Unadjusted | 0.27 | 0.11, 0.43 | <0.001 |
| Adjusted [a] | 0.15 | 0.04, 0.26 | 0.01 |
| *Lifetime number of children 1 to 4 (n = 390)* | | | |
| Unadjusted | 0.37 | 0.12, 0.62 | <0.01 |
| Adjusted [b] | 0.19 | 0.05, 0.34 | 0.02 |
| *Lifetime number of children 5 and above (n = 554)* | | | |
| Unadjusted | 0.21 | 0.003, 0.41 | 0.05 |
| Adjusted [b] | 0.11 | -0.07, 0.30 | 0.21 |

[a] Adjusted for mother's age and age squared, education (none, some primary, some secondary and more) and lifetime number of children (indicator variables)

[b] Adjusted for mother's age and age squared, education (none, some primary, some secondary and more) and with lifetime number of children (indicator variables), within each lifetime children subgroup

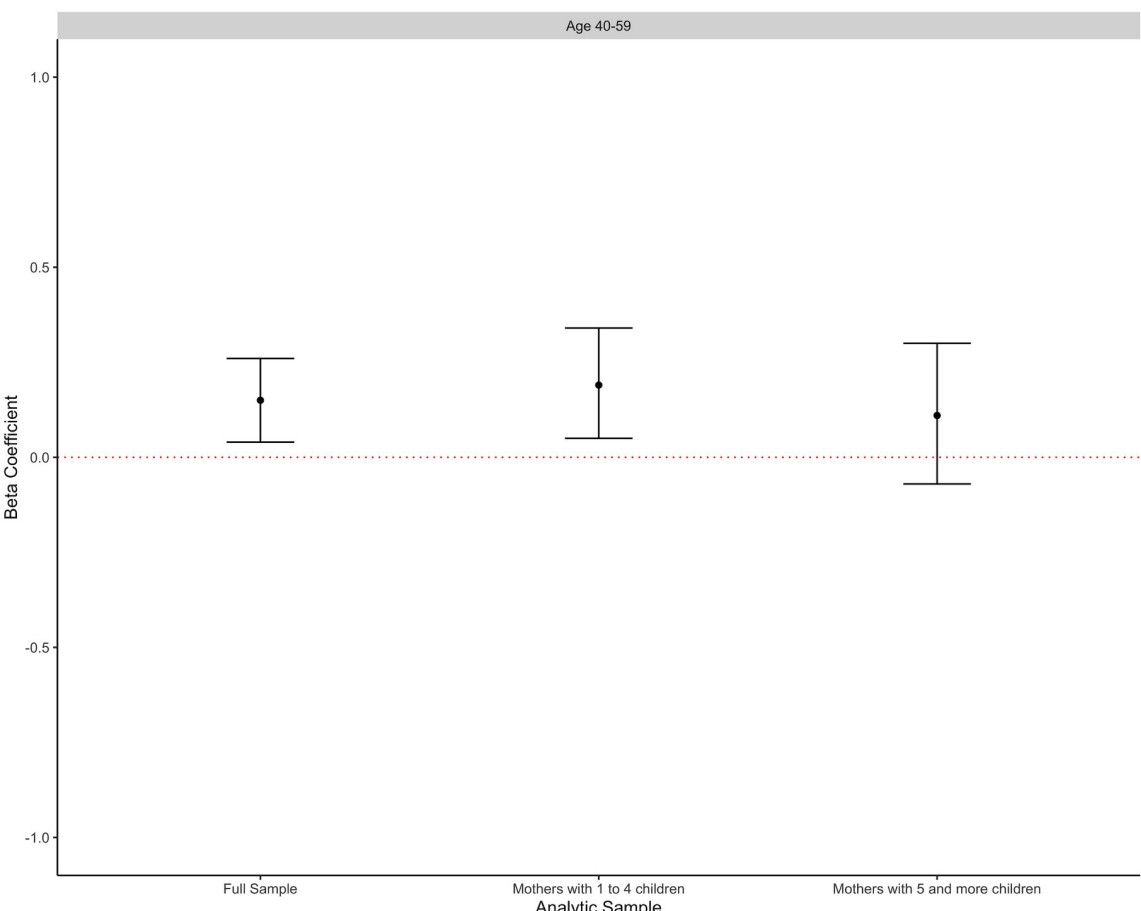

**Fig 2. Adjusted[a] estimates from ordinary least squares regression models of the cognitive function in child support grant (CSG)-eligible mothers who had high versus low duration of CSG eligibility in the three analytic samples.** [a]Adjusted for mother's age, mother's age squared, mother's education and lifetime number of children (indicator variables).

As a continuous variable, duration of CSG eligibility was not associated with cognitive function z-scores after adjusting for mother's age, education, and lifetime number of children (S1 Table in S1 File). When dichotomizing duration of CSG eligibility as ≤8 years ("low") versus >8 years ("high"), the association between duration of CSG eligibility and cognitive function z-scores for the full sample and subgroups were similar in magnitude and precision (β = 0.27; 95% CI: 0.14, 0.39; p value = 0.007, for mothers with 1–4 children) to the main analysis conducted with the 10-year cut-off (S2 Table in S1 File). When dichotomizing duration of CSG eligibility as ≤12 years ("low") versus >12 years ("high"), no association was observed between duration of CSG eligibility and cognitive function z scores in the full sample and in the lifetime number of children subgroups (β = 0.05; 95% CI: -0.09, 0.19; p value = 0.33, for mothers with 1–4 children) in adjusted analyses.

## Discussion

In this study of mothers in a rural, low-income South African population, we found that a higher duration of eligibility for Child Support Grant was associated with higher subsequent cognitive function z-scores among mothers between ages 40–59 years. We also found that in

mothers with one to four children in their lifetimes, CSG eligibility duration was associated with higher subsequent cognitive function z-scores. The magnitude of these associations was substantial. In our full sample, for every 5-year increase in age, the cognitive function z score lowered by an average of 0.18 SD units. Therefore, our study results suggest that over 10 years of cumulative eligibility for regular government cash transfers for child support in this group of mothers in their mid-life was equivalent to a nearly 5-year reduction in their cognitive ageing.

Only three other studies that we are aware of have used natural experiments to evaluate the causal effects of cash transfers on cognitive function in older adults [31–33]. In Mexico, cash transfers given to adults aged ≥70 years improved immediate and delayed recall scores with magnitudes that were equivalent to reductions in cognitive ageing of 5.5 and 12.4 years, respectively [31]. In the United States, cash transfers from social security income were associated with clinically meaningful improvements in multiple domains of cognitive function in older adults [32]. In South Africa, men who were exposed to expanded pension eligibility had higher cognitive function [33]. While our sample comprised women from a low-income, rural South African setting, our findings are consistent with the results from Mexico, United States and South Africa. We considered cognitive function across an earlier age range than had been examined in prior work by focusing on South African women in their mid-life. Cognitive decline can start during the middle years of adulthood and progressively worsen in later life [47, 48], potentially leading to ADRD [13]. Early detection of cognitive function changes in mid-life may help support interventions to delay or prevent ADRD onset.

A potential mechanism through which cash transfers may exert protective effects on mid-life cognitive function is via promoting cognitive reserve [17]. Accrual of cognitive reserve is thought to occur over the life course through cumulative cognitively stimulating experiences from childhood through adulthood [17, 49, 50]. People with high cognitive reserve are thought to be able to use compensatory behavioral or neural brain mechanisms to stave off the functional impacts of ageing-related brain pathology [17, 49]. In South Africa, CSG benefits account for nearly 40 per cent of income in the poorest recipient households and are accumulated over time in early- to mid-adulthood for mothers while they are raising children [21, 23]. Increased income, especially when sustained over time, can alleviate poverty, improve living conditions, reduce stress, and improve nutrition and health [22, 23], all of which may build cognitive reserve [51]. Prior research has shown that since the mothers were primary CSG recipients, they had a strong role in making financial decisions regarding the use of CSG benefits [23, 52]. It is possible that mother's economic empowerment within the household may be protective of their cognitive health. Furthermore, the CSG has been found to strengthen social support for the mothers via improved relationships with neighbors in the community and enabling them to participate in *stokvels*, which is a rotating savings and credit association [30]. High social support and social engagement is protective against cognitive ageing, particularly in adults in their mid-life [53, 54]. This may be due to them engaging in complex relationships and a feeling a sense of belonging and dignity in the community [55]. Another potential pathway to accrue cognitive reserve might have been expanding access to education in adolescent mothers who might still be in school at the time of CSG implementation. However, in our study sample, the youngest mothers were at least 24 years old in 1998, when CSG was first implemented. Therefore, their opportunities for expanding education might have closed by the time their children were CSG eligible. So, higher education levels might not plausibly explain the observed association between CSG and cognitive function in this group of mothers. Future research is needed to better understand the mechanisms through which improving the financial circumstances and independence of low-income middle-aged mothers may improve their cognitive health as they age.

We did not observe an association between higher duration of CSG eligibility and cognitive function in mothers with grand multiparity (with five or more children). Prior studies have found grand multiparity to be strongly associated with cognitive impairment and ADRD risk, which may overwhelm any protective effects of CSG eligibility for these women [56, 57]. Grand multiparity may result in decreased estrogen levels [58], long-term parenting stress [59], and increased risk of cardiovascular diseases [60], all of which can contribute to cognitive decline [59]. Our sample of rural South African women with a median of five lifetime number of children represented a relatively high fertility setting. There is scarce research on the association between parity and cognitive ageing in this setting and there may be unknown sociodemographic, socioeconomic, or psychosocial factors that mediate or moderate this relationship. Prior work using data from Agincourt HDSS found that the receipt of CSG had not motivated women to have more children [61]. Future studies are planned to investigate the relationships between reproductive history and cognitive ageing outcomes in the HAALSI cohort.

The findings from our sensitivity analysis suggest the use of the 10-year threshold may be adequate. As a continuous variable, duration of CSG eligibility was not associated with cognitive function. In South Africa, the inflation-adjusted food poverty line in April 2021 was 624 Rand (~33 USD) per person per month, while the CSG amount in April 2021 was only 480 Rand (~ 25 dollars) per child per month [62]. Therefore, the CSG benefit was of a magnitude smaller than that needed to cover the cost of the child's minimum required daily energy intake. However, CSG income is predominantly used to buy food for the child and the household [52], and sometimes is the only source of income in the households [63]. Therefore, it is plausible that a difference of one year of CSG eligibility is likely too little in duration and financial amounts to have had any meaningful impacts on the cognitive function of the mothers. We also tested the sensitivity of our threshold cut point at 10 years of CSG eligibility. When using the 8-year cut-off, our results were similar to the main analysis, suggesting that higher duration of CSG eligibility may be protective of the cognitive health of mothers with fewer children but not for those with grand multiparity. However, when we used the 12-year cut-off, we observed no association between CSG eligibility and cognitive function for mothers with 1–4 children, suggesting that the cut-off greater than 10 may be too high to detect any relationships between CSG eligibility and cognitive function. It is also possible that for mothers with 1–4 children, to be eligible for longer durations of CSG means having multiple children within short birth spacing intervals, which may neutralize any effect of CSG on mother's cognitive function.

## Limitations and strengths

Several aspects of the experimental design and data structure warrant careful interpretation of the results. We utilized duration of CSG eligibility and not actual CSG receipt in our analyses. The true relationship between CSG receipt and cognitive function may be different from our findings. However, given that not all mothers with CSG-eligible children applied for the grant [22], our exposed group included eligible mothers who did not receive the full CSG income we attributed to her. Therefore, our study findings plausibly yield more conservative estimates than we would expect if we used the actual amount of CSG received as our exposure. Furthermore, this method of exposure calculation has been previously used in South Africa to assess the relationship between years of Old Age Pension eligibility and health outcomes in older adults [64]. Cognitive function was measured at a single point in time, and we could not assess within-person cognitive change over time or incidence of cognitive impairment or ADRD. As the current cohort ages, studies using extended follow-up in HAALSI have been planned which will allow us to track cognitive change during ageing among the middle-aged women in

the current sample. Cognitive function was assessed using measures for orientation and epi-sodic memory, which does not include all possible domains of cognitive function. To address this issue, we ran a sensitivity analysis using cognitive function outcome that included numer-acy measures in addition to orientation, and episodic memory measures (S4 Table in S1 File). Our results were similar, with higher CSG eligibility duration associated with higher cognitive function scores in mothers with 1–4 children. Future studies should investigate other domains, such as executive function, language, or processing speed to gain a better understanding of whether and how cash transfers affect different domains of cognitive function. Longitudinal studies with large sample sizes and actual CSG receipt data are necessary to elucidate the rela-tionship between cash transfers and cognitive decline in South African women in their later-life. It is critical to know how long any cognitive health benefits of the CSG received in early-to mid-adulthood may last as beneficiaries age. It is also possible that pre-HAALSI baseline mortality patterns in women may have biased our findings, as aging cohorts are vulnerable to selective survival bias [65]. However, women who died prior to enrollment and women who were enrolled in HAALSI had very similar household asset indices (2.2 vs 2.3, respectively, from unpublished analyses using underlying vital statistics data collected by the Agincourt HDSS [20], covering the source population of the HAALSI cohort). Therefore, we believe that differences in household wealth between those enrolled in HAALSI and those not enrolled may not be large in magnitude to severely bias our results. That said, our findings should be interpreted as conditioned on survival until age 40 years and older in rural South Africa. Finally, our study findings may have limited generalizability outside of a rural South African setting.

This study has several strengths. We used data from a population-representative sample in a setting that is under-represented in the global dementia evidence base [34]. We used a quasi-randomized exposure variable for duration of CSG eligibility based on the CSG age-eligibility expansions and children's birth timings. This approach has also been used in previous research [28, 66]. This study data, which reports CSG eligibility duration since its inception until Janu-ary 2016, extends prior work on CSG receipt that have used self-reported data over a shorter follow-up period and/or used self-reported data to identify thresholds of CSG receipt duration [67, 68]. Since the increases in CSG amount over time was to match inflation and not to increase in value, the duration of CSG eligibility was considered an adequate proxy for CSG receipt. This approach allowed us to avoid recall or social desirability bias that can occur when self-reports are used or confounding by individual uptake patterns. This design strengthens causal inference for the association between CSG eligibility and mid- life cognitive function. We used validated and reliable measures of cognitive function, which are harmonized with those used in the US HRS and its International Partner Studies around the world and are meaningful for ADRD risk [43]. To the best of our knowledge, this is the first study examining the causal effects of cash transfer eligibility on mid- life cognitive function of mothers in a rural South African setting. Our findings bring evidence to policymakers for designing social protection plans that promote healthy cognitive ageing in mothers from rural, low-income settings.

## Conclusion

Our findings suggest that eligibility for cash transfers to support young children may protect the cognitive health of maternal recipients. This finding is critical since it indicates cash trans-fers can make promising interventions to protect cognitive health; and that CSG confers long-term cognitive health benefits to mothers in addition to children and households at the time of transfer. Future research should investigate how parity affects this relationship, how this

relationship may vary by the actual financial value of cumulative cash transfer receipt, and whether the protective effects of cash transfers like the CSG extend to later-life cognitive outcomes as beneficiaries age. Better understanding of the relationship between cash transfers and cognitive ageing is crucial, as this evidence would allow for policy recommendations, change in cost-benefit arguments and development of income supplementation programmes to promote healthy cognitive ageing in low-income settings.

## Supporting information

**S1 File. Child support grant variable construction and sensitivity analyses.**
(DOCX)

**S2 File. Mapping child support grant exposure.**
(XLS)

## Acknowledgments

We thank Dr. Audrey Pettifor and Erica Beidelman for their valuable contributions towards this manuscript.

## Author Contributions

**Conceptualization:** Molly Rosenberg.

**Data curation:** Rishika Chakraborty.

**Formal analysis:** Rishika Chakraborty.

**Funding acquisition:** Molly Rosenberg.

**Methodology:** Rishika Chakraborty, Janet Jock, Coady Wing.

**Supervision:** Molly Rosenberg.

**Validation:** Xiwei Chen.

**Visualization:** Xiwei Chen.

**Writing – original draft:** Rishika Chakraborty, Lindsay C. Kobayashi, Molly Rosenberg.

**Writing – review & editing:** Rishika Chakraborty, Lindsay C. Kobayashi, Janet Jock, Coady Wing, Meredith Phillips, Lisa Berkman, Kathleen Kahn, Chodziwadziwa Whiteson Kabudula, Molly Rosenberg.

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
