## [Decision Letter · Decision Letter 0]

27 Sep 2023

PONE-D-23-01857Child Support Grant expansion and cognitive function among women in rural South Africa: findings from a natural experiment in HAALSI cohortPLOS ONE

Dear Dr. Chakraborty,

Thank you for submitting your manuscript to PLOS ONE. After careful consideration, we feel that it has merit but does not fully meet PLOS ONE’s publication criteria as it currently stands. Therefore, we invite you to submit a revised version of the manuscript that addresses the points raised during the review process.

The authors should carefully read all comments and thoroughly revised the manuscript for further consideration.

We look forward to receiving your revised manuscript.

Kind regards,

Azizur Rahman, PhD

Academic Editor

PLOS ONE

Journal Requirements:

**Additional Editor Comments:**

The guidelines and style of PLOS ONE must be followed.

Reviewers' comments:

Reviewer's Responses to Questions

**Comments to the Author**

1. Is the manuscript technically sound, and do the data support the conclusions?

Reviewer #1: Yes

Reviewer #2: Partly

2. Has the statistical analysis been performed appropriately and rigorously? 

Reviewer #1: Yes

Reviewer #2: Yes

3. Have the authors made all data underlying the findings in their manuscript fully available?

Reviewer #1: Yes

Reviewer #2: No

4. Is the manuscript presented in an intelligible fashion and written in standard English?

Reviewer #1: Yes

Reviewer #2: Yes

5. Review Comments to the Author

Reviewer #1: This study aimed to exploit the exogenous variation in CSG eligibility, due to iterative age eligibility expansion policies, as a natural experiment to estimate the causal effect of cash transfers on cognitive function among biological mothers in South Africa. Good job! However, I have the following observations:

1. So far, I know PloS ONE has accepted structured abstracts. So, the authors need to revise the abstract.

2. In the Introduction section, it is suggested to update the statistic “In 2018, nearly 50 million people...”.

3. Add the strengths and limitations of the study as a separate section.

Reviewer #2: In this paper the authors use longitudinal data from the HAALSI cohort in Agincourt, South Africa to investigate the effect of expanded Child Support Grants on cognitive function. Ordinary least squares regression is used to model the relationship between cognitive function and duration of CSG eligibility, adjusted for a number of predictors including mother’s age, education and lifetime number of children.

In terms of the statistical analysis, the authors have tried several categorisations of the duration of CSG eligibility in order to find one that fits best. This suggests that the relationship is not strong to begin with and therefore the results in the paper are of waning interest. Also the fact that the best fitting model has to disregard the quantitative nature of duration so thoroughly means that the support for the hypothesis becomes weaker in practical terms.

I am concerned that the data are not strong enough to support the weighty hypothesis, which is that paying mothers more when their children are young can improve the cognitive function of the mothers later in life.

On the one hand, the proxy measure for cognitive function, a couple of simple short-term memory tests, seems likely to be weakly related to a fuller measure of cognitive function.

On the other hand, the difficulties in even measuring the amount of expanded CSG received by each mother seem likely to have a weak relationship with the actual amount of economic benefit accrued by each mother.

Also, it seems like an attribute such as wealth will be confounded with access to health services and therefore longevity, which could mean fewer women of lower socio-economic status are surviving long enough to exhibit symptoms of cognitive decline.

For these reasons, I am recommending rejection of the paper in its current form as its hypothesis is too stretched to be supported by the data available.

6. PLOS authors have the option to publish the peer review history of their article (what does this mean?). If published, this will include your full peer review and any attached files.

Reviewer #1: **Yes: **Md. Moyazzem Hossain

Reviewer #2: **Yes: **Alice Richardson

---

## [Author Response · Author response to Decision Letter 0]

5 Nov 2023

I have listed the changes made including the page number and line numbers [from the track changed document]

Reviewer 1

1: This study aimed to exploit the exogenous variation in CSG eligibility, due to iterative age eligibility expansion policies, as a natural experiment to estimate the causal effect of cash transfers on cognitive function among biological mothers in South Africa. Good job! However, I have the following observations.

Response: We thank the reviewer for their comments and suggestions. We have addressed them as described below.

2: So far, I know PloS ONE has accepted structured abstracts. So, the authors need to revise the abstract.

Response: We thank the reviewer for this observation. We have now structured the abstract under Background, Objectives, Methods, Results, and Conclusion sections (pages 2-3).

3.In the Introduction section, it is suggested to update the statistic “In 2018, nearly 50 million people…”.

Response: We have updated the statistic to a more recent one available for 2019 with projections for 2030, 2040, and 2050 (page 3, line 66-67): “In 2019, about 57 million people worldwide were living with dementia and this number is projected to increase to 83 million in 2030, 116 million in 2040 and 153 million by 2050 (Long et al., 2023; Nichols et al., 2022).” 

4. Add the strengths and limitations of the study as a separate section.

Response: We have now added a separate section under Discussion titled “Limitations and Strengths” below which we discuss study limitations and strengths (page 20, line 743).

Reviewer 2:

1. In this paper the authors use longitudinal data from the HAALSI cohort in Agincourt, South Africa to investigate the effect of expanded Child Support Grants on cognitive function. Ordinary least squares regression is used to model the relationship between cognitive function and duration of CSG eligibility, adjusted for a number of predictors including mother’s age, education and lifetime number of children. In terms of the statistical analysis, the authors have tried several categorisations of the duration of CSG eligibility in order to find one that fits best. This suggests that the relationship is not strong to begin with and therefore the results in the paper are of waning interest. Also, the fact that the best fitting model has to disregard the quantitative nature of duration so thoroughly means that the support for the hypothesis becomes weaker in practical terms.

Response: We thank the reviewer for this observation. We explored a threshold for CSG eligibility exposure in addition to operationalizing it continuously because if the threshold model more closely aligns with the true relationship then the continuous model would have obscured it and led to incorrect inference. In the continuous model, a 1-year increase in CSG, both in eligibility duration and in actual amount, may be too small to result in any meaningful cognitive benefit in the mothers. 

To identify whether a CSG eligibility threshold existed for cognitive function, we first visually graphed cognitive scores vs continuous CSG eligibility years, as shown in the supplementary materials. From this graph, we observed that mothers with 1- 10 years of CSG eligibility had similar level of cognitive function, but mothers with > 10 years of CSG eligibility showed noticeably higher cognitive function scores (S3 Fig). Thus, we selected 10 years as our threshold. However, to further test the robustness of this initial threshold we conducted two sensitivity analyses using alternate cut-offs: one at 8 years and the other at 12 years. We found that at the 8-year threshold, there was a stronger protective effect on cognitive function (β = 0.27; 95% CI: 0.14, 0.39; p value = 0.007) compared to the 10-year cut-off used in main analysis (β = 0.19; 95% CI: 0.05, 0.34; p value = 0.02) for mothers with 1-4 kids. However, at 12-year cut-off, we found no association between CSG eligibility and cognitive function for mothers with 1-4 kids (β = 0.05; 95% CI: -0.09, 0.19; p value = 0.33). Thus, our findings seem to suggest that there exists a threshold for the relationship between cumulative CSG eligibility duration and cognitive function, particularly for mothers with 1-4 kids. However, beyond this threshold, that relationship weakens. It is possible that cut-off beyond 10 years is too high to detect differences. It is also possible that mothers with 1 - 4 children with > 10 years of CSG eligibility had smaller birth spacing between children and the cognitively taxing process of child rearing diluted the effect of CSG eligibility and cognitive function. There appears to be complex interplay between CSG, parity, and cognitive function which we are exploring in future studies.

To contextualize this cash benefit, in South Africa, the inflation-adjusted food poverty line in April 2021 was 624 Rand (~33 USD) per person per month, however the CSG benefit amount in April 2021 was only 480 Rand (~25 USD) per child per month. Therefore, the CSG amount was of a magnitude smaller than that needed to cover the cost of the child’s minimum required daily energy intake. However, we know that CSG was predominantly used to buy food for the child and the household (DSD et al., 2011), and often was the only source of income in the households (Luthuli et al., 2022). Therefore, while the CSG helps households alleviate their acute food insecurity, we hypothesize that it will likely take a longer duration of cumulative eligibility to help build cognitive reserve that can protect cognitive function. Establishing a threshold for assessing impact of CSG duration on child outcomes have been used in prior studies (Zembe-Mkabile et al., 2016). For these reasons, we feel confident that a CSG threshold was reasonable to expect for maternal cognitive outcomes. 

We have clarified this point in our Discussion section (page 19, line 714- 741) “As a continuous variable, duration of CSG eligibility was not associated with cognitive function. In South Africa, the inflation-adjusted food poverty line in April 2021 was 624 Rand (~33 USD) per person per month, while the CSG amount in April 2021 was only 480 Rand (~ 25 dollars) per child per month (Statistics South Africa, 2022). Therefore, the CSG benefit was of a magnitude smaller than that needed to cover the cost of the child’s minimum required daily energy intake. However, CSG income is predominantly used to buy food for the child and the household (DSD et al., 2011), and sometimes is the only source of income in the households (Luthuli et al., 2022). Therefore, it is plausible that a difference of one year of CSG eligibility is likely too little in duration and financial amounts to have had any meaningful impacts on the cognitive function of the mothers. We also tested the sensitivity of our threshold cut point at 10 years of CSG eligibility. When using the 8-year cut-off, our results were similar to those in the main analysis, suggesting that higher duration of CSG eligibility may be protective of the cognitive health of mothers with fewer children but not for those with grand multiparity. However, when we used the 12-year cut-off, we observed no association between CSG eligibility and cognitive function for mothers with 1-4 kids, suggesting that the cut-off greater than 10 may be too high to detect any relationships between CSG eligibility and cognitive function. It is also possible that for mothers with 1 -4 children, to be eligible for longer durations of CSG means having multiple children within short birth spacing intervals, which may neutralize any effect of CSG on mother’s cognitive function.” 

2. On the one hand, the proxy measure for cognitive function, a couple of simple short-term memory tests, seems likely to be weakly related to a fuller measure of cognitive function.

Response: We agree with the reviewer that mental status (orientation measures) and memory status (immediate and delayed recall measures) used to create the cognitive function outcome variable do not capture the full cognitive function of an individual. To address this limitation, we did an additional sensitivity analysis using a different cognitive function outcome variable which includes a numeracy measure (forward count and number skip pattern) in addition to orientation, immediate, and delayed recall measures. It yielded similar results, with higher CSG eligibility duration conferring a protective effect on cognitive function in mothers with 1- 4 children (β = 0.17; 95% CI: 0.02, 0.30; p value = 0.03). 

Despite the limitations in the cognitive function measure used in the study, this cognitive battery has been derived from the United States Health and Retirement Study (HRS) screening test for dementia and has been validated and harmonized with other cognitive measures used in US HRS and its International Partner Studies which includes HAALSI (Kobayashi et al., 2020). Furthermore, episodic memory deficits are hallmark symptoms of ADRD, are sensitive to aging-related changes, and can be detected in preclinical ADRD stages (Bäckman et al., 2001; Salthouse, 2009). This cognitive variable has also been used in previous studies (Jock et al., 2023; Kobayashi et al., 2017; Kobayashi et al., 2019). 

However, like some prior studies (Kobayashi et al., 2019), we refrain from using the numeracy measure in the main analysis because it is a skill learned through education. Given the limited educational opportunities that our study sample faced during Apartheid, those who are not numerate may be particularly disadvantaged for the numeracy tests that are not reflective of overall cognitive function or decline. 

We have further clarified this limitation in our Limitation and Strengths section (page 21, line 774 - 781): “Cognitive function was assessed using measures for orientation and episodic memory, which does not include all possible domains of cognitive function. To address this issue, we ran a sensitivity analysis using cognitive function outcome that included numeracy measures in addition to orientation, and episodic memory measures (Supplementary Materials, S4 Table). Our results were similar, with higher CSG eligibility duration associated with higher cognitive function scores in mothers with 1- 4 children. Future studies should investigate other domains, such as executive function, language, or processing speed to gain a better understanding of whether and how cash transfers affect different domains of cognitive function.”

We have also clarified the use of numeracy measures in our Methods section (page 11, line 463- 472): “These measures were selected for use as the inability to orient oneself in time and place and loss of episodic memory are hallmark symptoms of ADRD, and are highly sensitive to ageing-related change (Bäckman et al., 2001; Salthouse, 2009). This cognitive battery was adapted from the United States Health and Retirement Study (US HRS) and has been used in other HRS International Partner Studies around the world (Kobayashi et al., 2020; Wu et al., 2013) including South Africa (Jock et al., 2023; Kobayashi et al., 2017; Kobayashi et al., 2019).The cognitive battery also had 2 numeracy measures (forward count and number skip pattern), which we did not include in the present analysis. This is because numeracy is a skill learned through education and given the limited educational opportunities that our study sample faced during Apartheid, those who are not numerate may be particularly disadvantaged for the numeracy tests (Kobayashi et al., 2019).” 

3. On the other hand, the difficulties in even measuring the amount of expanded CSG received by each mother seem likely to have a weak relationship with the actual amount of economic benefit accrued by each mother.

Response: It is likely that the relationship between the duration of CSG eligibility and cognitive function in our study may be more conservative compared to the relationship between actual CSG receipt and cognitive function. A recent study showed that about 83% of all CSG -eligible children received the grant. Although there is likely relatively high concordance between eligibility and uptake, the discordance could be explained by barriers in accessing CSG and/or misinformation about the CSG eligibility. Thus, our CSG-eligibility exposure may include mothers who were eligible but did not receive the CSG. Therefore, our study estimates are likely more conservative than if we used the actual amount of CSG received as our exposure. It is likely that we would observe a stronger relationship between actual amount of CSG and cognitive function.

Furthermore, women of similar age and similar number of children may have varying CSG eligibility durations (based on childbirth timing) due to CSG age eligibility expansions. Thus, in this study we leverage the exogenous variation in CSG eligibility over time. In contrast, the actual uptake of CSG benefits may be associated with individual risk factors or early outcomes such as education or high early life cognitive functioning that could confound the relationship between CSG and later outcomes. Therefore, we focus on the “intent to treat” relationship between exogenous variation in years of CSG eligibility and cognitive functioning, which yields policy relevant findings. Moreover, this method of calculating years of cash transfer eligibility was used in a recent study to assess the relationship between Old Age Pension and health outcomes in South Africa (Riumallo Herl et al., 2022). 

We have clarified this point in our Limitations and Strengths section (page 20, line 747 - 754): “We utilized duration of CSG eligibility and not actual CSG amount in our analyses. Therefore, the true relationship between CSG amount and cognitive function may be different from our findings. However, given that not all mothers with CSG-eligible children applied for the grant (DSD et al., 2012), our exposed group included eligible mothers who did not receive the full CSG income we attributed to her. Therefore, our study findings plausibly yield more conservative estimates than we would expect if we used the actual amount of CSG received as our exposure. Furthermore, this method of exposure calculation has been previously used in South Africa to assess the relationship between years of Old Age Pension eligibility and health outcomes in older adults (Riumallo Herl et al., 2022).” 

4. I am concerned that the data are not strong enough to support the weighty hypothesis, which is that paying mothers more when their children are young can improve the cognitive function of the mothers later in life.

Response: We have revised our paper carefully to increase transparency about our data limitations. We would also like to note a strength of our approach relative to approaches represented in the existing literature. We calculated CSG eligibility since its inception in April 1998 to January 2016 and covered an 18-year exposure period. This extends prior studies that have used self-reported CSG receipt data, which may be biased, at one or two timepoints and established thresholds from self-reported data to create CSG exposure variables (Cluver et al., 2013; Zembe-Mkabile et al., 2016). Further, our cognitive measures have been adapted from the US HRS and are reliable and valid for this study setting and harmonized with those used in HRS and its International Partner Studies (Kobayashi et al., 2020). 

We acknowledge the above in our Limitations and Strengths section (page 22, line 806-815): “This study data, which reports CSG eligibility duration since its inception until 2016, extends prior work on CSG receipt that have used self-reported data over a shorter follow-up period and/or used self-reported data to identify thresholds of CSG receipt duration (Cluver et al., 2013; Zembe-Mkabile et al., 2016). Since the increases in CSG amount over time was to match inflation and not to increase in value, the duration of CSG eligibility was considered an adequate proxy for CSG receipt. This approach allowed us to avoid recall or social desirability bias that can occur when self-reports are used or confounding by individual uptake patterns. This design strengthens causal inference for the association between CSG eligibility and mid- life cognitive function. We used validated and reliable measures of cognitive function, which are harmonized with those used in the US HRS and its International Partner Studies around the world and are meaningful for ADRD risk(Kobayashi et al., 2020).”

We have also expanded our limitation section, as described above (page 20, line 746- 794): “We utilized duration of CSG eligibility and not actual CSG receipt in our analyses… Cognitive function was assessed using measures for orientation and episodic memory, which does not include all possible domains of cognitive function. To address this issue, we ran a sensitivity analysis using cognitive function outcome that included numeracy measures in addition to orientation, and episodic memory measures… It is also possible that pre-HAALSI baseline mortality patterns in women may have biased our findings, as aging cohorts are vulnerable to selective survival bias… our findings should be interpreted as conditioned on survival until age 40 years and older in rural South Africa. Finally, our study findings may have limited generalizability outside of a rural South African setting.”

5. Also, it seems like an attribute such as wealth will be confounded with access to health services and therefore longevity, which could mean fewer women of lower socio-economic status are surviving long enough to exhibit symptoms of cognitive decline.

Response: We very much agree with this point - only those who survived until at least 40 years of age were included in HAALSI. Therefore, those who died prior to this, will not be represented in this study. Aging research is particularly vulnerable to selective survival bias (Weuve et al., 2015). It is possible that those who died prior to enrollment in HAALSI had different socioeconomic status than those who were enrolled. Interestingly, authors of this paper have also investigated the differences in sociodemographic variables between those with and without pre-baseline mortality in HAALSI (unpublished manuscript). This was possible with underlying vital statistics data on the source population of the HAALSI cohort. We found that those who died prior to enrollment had only slightly lower household asset index scores compared to those who were enrolled (2.2 vs 2.3, respectively). Therefore, wealth may not strongly confound our findings. However, our study conclusions are likely restricted to the current HAALSI population and measures cognitive function at baseline only. 

We have added this limitation in our Limitation and Strengths section (page 21, Line 785 -793): “It is possible that pre-HAALSI baseline mortality patterns in women may have biased our findings, as aging cohorts are vulnerable to selective survival bias (Weuve et al., 2015) . However, women who died prior to enrollment and women who were enrolled in HAALSI had very similar household asset indices (2.2 vs 2.3, respectively, from unpublished analyses using underlying vital statistics data collected by the Agincourt HDSS (Kahn et al., 2012), covering the source population of the HAALSI cohort). Therefore, we believe that differences in household wealth between those enrolled in HAALSI and those not enrolled may not be large in magnitude to severely bias our results. That said, our findings should be interpreted as conditioned on survival until age 40 years and older in rural South Africa.” 

6. For these reasons, I am recommending rejection of the paper in its current form as its hypothesis is too stretched to be supported by the data available.

Response: Thank you for your valuable comments. We have incorporated your feedback and suggestions and believe that this has strengthened our manuscript. We hope that we have adequately addressed your concerns.

7. Have the authors made all data underlying the findings in their manuscript fully available? - No

Response: The HAALSI baseline data used in this study is publicly available in the Harvard Dataverse. We have clarified this in our Data Availability Statement (page 23, line 841): The HAALSI data used in this article are publicly available as “HAALSI Baseline Survey” here: https://dataverse.harvard.edu/dataverse/haalsi

Editor 

1. The guidelines and style of PLOS ONE must be followed.

Response: We have revised the manuscript to ensure that it follows the guidelines and style of PLOS ONE.

---

## [Decision Letter · Decision Letter 1]

11 Jan 2024

Child Support Grant expansion and cognitive function among women in rural South Africa: findings from a natural experiment in the HAALSI cohort

PONE-D-23-01857R1

Dear Dr. Chakraborty,

We’re pleased to inform you that your manuscript has been judged scientifically suitable for publication and will be formally accepted for publication once it meets all outstanding technical requirements.

Within one week, you’ll receive an e-mail detailing the required amendments. When these have been addressed, you’ll receive a formal acceptance letter, and your manuscript will be scheduled for publication.

Kind regards,

Azizur Rahman, PhD

Academic Editor

PLOS ONE

Additional Editor Comments (optional):

1) The authors have done a careful revision and adequately addressed all comments.

2) They must meet the journal's technical requirements (or if anything is pending).

Congratulations and thanks for the authors' sincere efforts.

Reviewers' comments:

Reviewer's Responses to Questions

**Comments to the Author**

1. If the authors have adequately addressed your comments raised in a previous round of review and you feel that this manuscript is now acceptable for publication, you may indicate that here to bypass the “Comments to the Author” section, enter your conflict of interest statement in the “Confidential to Editor” section, and submit your "Accept" recommendation.

Reviewer #1: All comments have been addressed

Reviewer #2: (No Response)

2. Is the manuscript technically sound, and do the data support the conclusions?

Reviewer #1: (No Response)

Reviewer #2: Yes

3. Has the statistical analysis been performed appropriately and rigorously? 

Reviewer #1: (No Response)

Reviewer #2: Yes

4. Have the authors made all data underlying the findings in their manuscript fully available?

Reviewer #1: (No Response)

Reviewer #2: No

5. Is the manuscript presented in an intelligible fashion and written in standard English?

Reviewer #1: (No Response)

Reviewer #2: Yes

6. Review Comments to the Author

Reviewer #1: (No Response)

Reviewer #2: In this paper the authors use longitudinal data from the HAALSI cohort in Agincourt, South Africa to investigate the effect of expanded Child Support Grants on cognitive function. Ordinary least squares regression is used to model the relationship between cognitive function and duration of CSG eligibility, adjusted for a number of predictors including mother’s age, education and lifetime number of children. My review raised a number of concerns around the strength of the data and whether it was sufficient to inform the far-reaching hypothesis, that a child support grant can affect the cognitive function of the child’s carers in later life.

The authors have addressed all the concerns that I raised and considerably added to the paper to explain and justify their results. 

7. PLOS authors have the option to publish the peer review history of their article (what does this mean?). If published, this will include your full peer review and any attached files.

Reviewer #1: No

Reviewer #2: No

---

## [Editor Report · Acceptance letter]

12 Feb 2024

PONE-D-23-01857R1 

PLOS ONE

Dear Dr. Chakraborty, 

I'm pleased to inform you that your manuscript has been deemed suitable for publication in PLOS ONE. Congratulations! Your manuscript is now being handed over to our production team.

Kind regards, 

on behalf of

Professor Azizur Rahman 

Academic Editor

PLOS ONE